# Phonemic composition influences words' aesthetic appeal and memorability

Theresa Matzinger [1,2‡]*, David Košić[1‡]

1 Department of English and American Studies, University of Vienna, Vienna, Austria, 2 Vienna Cognitive Science Hub, University of Vienna, Vienna, Austria

‡ joint first authors
* theresa.matzinger@univie.ac.at

## Abstract

Positive emotional responses from pleasant experiences are known to enhance memory, yet the relationship between aesthetic appeal and linguistic memory remains understudied. To investigate this relationship, we designed pseudowords of varying appeal based on Crystal's [1] phoneme rankings. Native English-speaking participants actively memorized these pseudowords and completed a free recall test, followed by two rounds of appeal ratings. Our results showed that, contrary to our predictions, pseudowords designed to be of intermediate appeal were rated as more appealing than those designed to be highly appealing or unappealing. Nevertheless, pseudowords designed to be highly appealing were recalled most frequently – even though participants themselves did not rate them as highly appealing. Also, overall, recalled words received higher appeal ratings from participants than non-recalled ones. These findings suggest that the phonemic and phonotactic composition of words may, indeed, have aesthetic value that correlates with words' memorability. This encourages further exploration into how appeal interacts with other factors influencing linguistic cognition, including occurrence frequency or complexity. Our findings can inform applications in language learning, teaching, and marketing, while also offering theoretical contributions to our understanding of language evolution and change.

## 1. Introduction

Traditionally, aesthetics has been regarded as a philosophical discipline dedicated to studying the artistic value of cultural artifacts [1,2]. However, recent studies suggest that aesthetic appeal is not limited to art but can also be observed in mundane everyday experiences [2–4] such as the design of everyday objects [5], landscapes [6], and the sounds of spoken language [7]. This broader understanding of aesthetics led to the emergence of the field of phonaesthetics, which is dedicated to studying the aesthetic properties of speech sounds [8]. Although the idea of phonaesthetics

**Data availability statement:** We preregistered all hypotheses, study protocols, and analyses, including provisional R files, on the Open Science Framework (Preregistration date: 27 June, 2023; https://osf.io/npe2g/overview). All data and R files used for the analysis are available in the Open Science Framework repository and can be accessed at https://osf.io/npe2g/overview.

**Funding:** This work was supported by a Disruptive Innovation Grant from the Austrian Academy of Sciences and the Austrian Science Fund (grant number: DI_2023-108_MATZINGER_BEALP) awarded to Theresa Matzinger. The funders had no role in study design, data collection and analysis, decision to publish, or preparation of the manuscript.

**Competing interests:** The authors have declared that no competing interests exist.

dates back to the 1920s [9], it has since received little scholarly attention. Recently, research has started to focus on the aesthetic appeal of languages as such [10–12], but systematic studies on the aesthetic appeal of individual phonemes and phonotactic patterns remain scarce, cf. [8].

This is surprising, since understanding if and how phonological features differ in their aesthetic appeal may have implications for several fields and disciplines. For example, it has sociolinguistic relevance because it can help to explain why certain languages are perceived as more beautiful than others [11,12] and might in turn help to understand and counteract language-based prejudices. Further, it might have implications for language learning or language acquisition and may influence how frequently certain words are used. This may in turn also affect, for example, language change [13]. To draw meaningful conclusions for those fields, it is essential to first establish whether linguistic patterns vary in their aesthetic appeal at all, and if so, in what ways they vary. Therefore, the main objective of our study is to determine which individual phonemes and phonotactic patterns are perceived as more aesthetically appealing than others. A secondary goal is to explore how these aesthetic preferences may interact with other cognitive processes, such as memory, thereby expanding the scope of phonaesthetic research to encompass more complex cognitive phenomena.

The idea that individual phonemes and phonotactic patterns may differ in their aesthetic appeal finds support in the extensive literature on sound symbolism. This body of research shows that speech sounds are systematically associated with perceptual and conceptual domains such as size [14,15], shape [16–18], color [19], or body parts [20]. Cross-linguistic and large-scale studies further suggest that such associations are not limited to a few isolated examples but reflect widespread tendencies [20] and also play a role in children's language acquisition strategies [21,22]. Taken together, this research highlights that sounds themselves can carry meaning and evoke affective responses. In this sense, it is plausible that certain phonological patterns are linked to positive or negative emotional values, and thus to aesthetic appeal. This perspective provides a theoretical foundation for our study, which seeks to determine whether sound symbolic associations extend to perceptions of linguistic appeal.

One of the few early studies that addressed the issue of aesthetic appeal of phonemes and other linguistic features is by Crystal [8]. His analysis involved compiling lists of words typically regarded as appealing, examining their phonemic makeup, and comparing the frequency of phonemes in appealing words to their frequency in conversational English. He found a tendency for appealing words to contain continuants (e.g.,/l/,/s/), nasals (e.g.,/m/,/n/), front vowels (e.g.,/i/,/e/), and diphthongs (e.g.,/aɪ/,/eɪ/). Conversely, Crystal noted that appealing words contained fewer "harsh" or abrupt sounds, such as plosives (e.g.,/p/,/g/), postalveolar fricatives (e.g.,/ʃ/,/ʒ/), and affricates (e.g.,/tʃ/,/dʒ/). While Crystal's study is an important first step in investigating the appeal of linguistic patterns, his study also comes with many shortcomings. For example, Crystal's observations were based on subjective lists of aesthetically pleasing words compiled from poets, lexicographers, and public polls, which reflected

individual preferences and biases rather than being derived from a controlled, systematic analysis. Also, the lack of control for the semantic content of the words in question may compromise potential conclusions about the inherent aesthetic qualities of the linguistic features investigated, as the words' meanings may have affected their aesthetic perception. Thus, overall, to gain a more reliable understanding of the aesthetic appeal of linguistic patterns, more comprehensive and systematic investigations are needed.

Though not explicitly focusing on *aesthetic* appeal, other studies that examined closely related qualities such as valence and preference support the idea that different sounds and sound patterns may also differ in their aesthetic appeal. For instance, Pogacar et al. [23] found that top brand names often featured specific sounds such as front vowels, suggesting a link between phonemic quality and consumer preference. Similarly, a study by Mooshammer et al. [24] that examined conlangs found that fricatives were consistently associated with positive meanings, while stops like/k/ and/g/ were linked to negative connotations. This is further supported by Lev-Ari & McKay [25], who found that swear words are significantly less likely to contain approximants, suggesting a systematic avoidance of soft and pleasant sounds in negatively charged vocabulary. In line with this, studies of profanity and emotionally loaded vocabulary have shown that plosives play a key role: Yardy [26] observed that English swear words contain proportionally more plosives (e.g.,/k/,/t/) than genres with positive associations such as carols and lullabies, which instead favor more sonorant consonants (e.g.,/l/,/m/,/n/,/w/). Bergen [27] likewise noted that English monosyllabic swear words tend to end in plosives more often than non-swear controls, and Reilly et al. [28] found that both existing and novel taboo words were rated as more taboo when they contained plosives. Also, Aryani et al. [7] suggest that speech sounds are linked to affective meaning and found that words with short vowels, voiceless consonants and hissing sibilants such as/s/ are perceived as more negative and arousing. Finally, also, Körner & Rummer [29] found that, across languages, words containing/i/ were matched with positively connotated people and objects, while words containing/o/ or/u/ were used for negatively connotated ones. This matches iconicity studies from consumer research showing that brand names containing front vowels like/i/ were associated with positive traits such as lightness and smallness, while names with back vowels like/o/ or/u/ conveyed traits like heaviness, illustrating how phonemic qualities influence people's perception of products [30]. Taken together, these studies, which show links between phonemic quality and valence, suggest that there might also be a link between phonemic quality and aesthetic appeal. However, it remains unclear if individual phonemes that differ in valence, will differ in appeal in a similar way, and how this will influence the general appeal of the words they constitute.While there is already little systematic research on the aesthetic appeal of phonemes and phonotactic patterns, there is even less research linking the appeal of speech parameters to memory. One study indirectly addressing this topic is Matzinger et al. [13], who found that the aesthetic appeal of prosodic patterns correlates with their usefulness for segmenting continuous speech into individual words. More specifically, they found that trisyllabic pseudowords with a lengthened final syllable were rated as more beautiful and likable than those with a shortened final syllable. This preference aligned with participants' ability to more accurately identify the same words in a continuous speech stream when the final syllable was lengthened rather than shortened [31]. Although Matzinger et al.'s [13] study is not directly focusing on memory, it suggests that aesthetically pleasing prosody may facilitate language processing and learning.

Moreover, despite limited research on direct links between phonaesthetics and memory, there is abundant evidence from other domains such as music and vision that suggest that aesthetic appeal can positively impact human memory. Aesthetically appealing stimuli generally trigger the human reward system [32,33], prompting humans to seek gratification from these stimuli and devote their attention to them over others that may be considered less appealing or, functionally, less rewarding [34,35]. This heightened attention demands high cognitive involvement, ultimately enhancing the likelihood of memory formation [36]. Furthermore, the cascade of positive emotions and heightened arousal associated with appealing stimuli may contribute to memory consolidation, ensuring these newly formed memories are stored long-term [32,33]. Even when humans do not actively seek or perceive rewarding stimuli, they are still more likely to be memorized: arousing stimuli are better remembered even if processed incidentally, i.e., without focused attention [37]. Conversely, less

appealing or "boring" stimuli significantly decrease memory performance and are likely to be quickly forgotten [38]. As a result, if speech sounds or phonotactic patterns differ in their aesthetic appeal, these differences in emotional arousal may have significant implications for our understanding of phonological short-term memory [39] functioning in the context of language acquisition.

While the majority of evidence suggests that positive emotions boost memory, there are some studies that claim otherwise. For example, research on negativity bias suggests that negative experiences elicit more powerful emotions than neutral and positive experiences, leading to more in-depth processing and faster encoding, consolidation, and retrieval of information [40,41]. Seen from an evolutionary standpoint, this tendency to retain negative experiences likely served an adaptive function since remembering threats and dangers would be considered more critical than retaining positive memories [42]. However, some scholars argue that both positive and negative emotions enhance memory and cognitive processing through similar mechanisms, emphasizing the role of emotional arousal as opposed to valence [43,44]. Thus, if phonaesthetic appeal affects word memory, it remains uncertain whether pleasing or displeasing sounds will enhance recall more effectively.

To provide a more comprehensive picture of potential links between phonaesthetics and memory, the aims of our study were twofold: (a) to test the aesthetic appeal of phonemes and phonotactic patterns, thereby providing a more objective validation of Crystal's ratings, and (b) to examine whether this appeal correlates with word memorability. For this reason, we conducted a pseudoword learning experiment, in which participants were exposed to pseudowords that were designed to be appealing, neutral, or unappealing, as per Crystal [8]. The experiment consisted of a rating task, in which participants had to judge how appealing they found the pseudowords, and a free recall test, in which they had to recall the pseudowords they had memorized. We used this data to validate and update Crystal's proposed rankings of phonemic appeal, and to correlate the word ratings with their recall frequency. While our focus is on word memory in general, rather than on a strict distinction between first (L1) and second language (L2) word learning, the use of English-inspired pseudowords and native English-speaking participants (see below) means that our study draws on both L1 phonotactic familiarity and L2-like learning conditions, potentially bearing relevance for both L1 and L2 learning contexts.

We hypothesized that we would be able to confirm Crystal's ratings and predicted that pseudowords containing sounds that ranked highly in Crystal's hierarchy (from now on called "appealing words" or the "appealing condition") would receive higher ratings of aesthetic appeal than words containing sounds that ranked lowly in Crystal's hierarchy (from now on called "unappealing words" or the "unappealing condition"), with pseudowords containing sounds from the middle of Crystal's hierarchy (from now on called "neutral words" or the "neutral condition") receiving intermediate ratings.

Regarding memory, we hypothesized that appeal would influence word memorization. Specifically, we predicted that, if high appeal positively influences memory, words from the appealing condition should be recalled most frequently and, conversely, words from the unappealing condition should be recalled least frequently. Words from the neutral condition should fall somewhere in between. Alternatively, if unappealing words are more memorable, we would expect the opposite effect. If the extremes of the emotional scale, irrespective of their valence, are crucial for memory formation, appealing and unappealing pseudowords should be recalled equally often but more often than neutral stimuli.

## 2. Methods

### 2.1. Experimental conditions, procedure and setting

To test whether the phonemic makeup of words influenced their perceived aesthetic appeal, and whether this influenced the words' memorability, we conducted an experiment that was divided into three phases: stimulus presentation, stimulus recall, and stimulus rating (Fig 1). In the first phase, the stimulus presentation phase, participants were exposed to a set of stimuli that were designed to be either appealing, neutral or unappealing (based on Crystal [8]; see Section 3.3. for more details). Participants were exposed to four stimuli in each of the three conditions visually and audibly. Note that visual

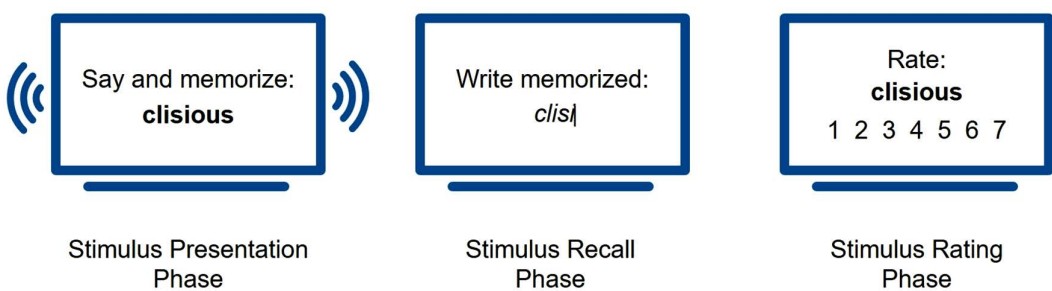

**Fig 1. Experimental setup.** The experiment consisted of three phases: (1) *Stimulus Presentation*, where participants memorized visually and audibly presented stimuli; (2) *Stimulus Recall*, where they typed the stimuli that they recalled; and (3) *Stimulus Rating*, where they rated each stimulus on a 7-point Likert scale for appeal.

presentation of the stimuli may confound phonological memory, especially when there are grapheme-to-phoneme ambiguities or mismatches between auditory and orthographic representations. We still opted for a visual exposure to the stimuli because pilot studies showed that purely auditory exposure led to very low recall performances. Presenting the spelling alongside the auditory form during the learning phase allowed us to increase task sensitivity and capture meaningful differences between conditions. The auditory input ensured that the orthographic representations were interpreted similarly across participants.

The stimuli were displayed on the screen in random order, with each stimulus appearing six times for three seconds, as repeated exposure has been shown to enhance associative memory [45,46]. Audio recordings of the stimuli's pronunciation played automatically in the background, with the audio starting immediately as a written stimulus appeared on the screen. Participants were instructed to orally repeat each stimulus to ensure full concentration on the task and further promote memorability. The visual display of each stimulus for three seconds provided enough time for participants to listen to the stimulus, to orally repeat it once, and to have a brief buffer time before the start of the next stimulus. At the same time, this interval was too short for participants to take notes or use other memory aids. To verify that participants did not use external memory aids, we examined recall performance for outliers. There were no cases of (near-)perfect performance, which makes note-taking highly unlikely. After all stimuli had been presented, the setup automatically proceeded to the second phase, the stimulus recall phase. Participants could take a self-timed break before starting with the stimulus recall phase.

In the stimulus recall phase, participants were prompted to recall and type the words that they had memorized. They were informed that they should enter all words that they had remembered, even if they were not fully certain about their answers. There was no time limit for participant responses during the stimulus recall phase. Participants could recall the words they remembered at their own pace, as a strict time constraint might have added unnecessary pressure, potentially affecting recall performance negatively.

In the third phase, the stimulus rating phase, participants were instructed to rate the appeal of each stimulus on a 7-point Likert scale ranging from "unappealing" to "appealing". There were two separate rating rounds, i.e., each stimulus was presented twice to check for consistency of the ratings within individuals. Within the two rating rounds, the stimuli were displayed in random order. Participants gave their ratings by pressing a corresponding number key on the keyboard. To elicit the participants' ratings, we used the prompt "Rate the following word's appeal". We decided to not further instruct or brief the participants on the term "appeal" because, even after providing definitions, it would remain a very subjective term that is prone to individual interpretations [47,48], and we intended to elicit intuitive spontaneous responses rather than responses biased or superimposed by potential instructions. After completing the ratings, the participants were debriefed, thanked for their participation and the experiment was concluded.

The experiment was conducted online, and participants were instructed to participate from a quiet room without distractions. Participants were also required to use headphones to ensure proper presentation of auditory stimuli.

## 2.2. Participants

The study comprised a sample of 100 native English speakers (47 male, 48 female, 5 other; mean age 29.35±SD 10.21 years). We did not make any restrictions regarding the variety of English spoken. The choice of our sample size was based on the results of a power analysis (using the *pwr* package in R [49]), with a desired power of 0.8, a significance level of 0.05 and an assumed small effect size [50]. Participants were recruited through the crowdsourcing platform Prolific (www.prolific.com, 2024) and received a remuneration of £ 1.0 for their approximate 10-minute participation. The research protocol received approval from the Scientific Research Ethics Committee at the University of Vienna (reference number: 00998) and all participants provided informed consent in adherence to the declaration of Helsinki.

## 2.3. Stimulus design

We designed twelve English-sounding pseudowords to use as stimuli in our study. We opted for pseudowords that did not resemble real English words to prevent the words' appeal from being influenced by semantic content or connotations. Likewise, the fact that all included sounds were part of the English phoneme inventory was intended to prevent negative effects on appeal due to phonological unfamiliarity. All pseudowords were trisyllabic and followed an identical structure: they started with a consonant cluster, followed by a vowel and single consonant, and ended in the suffix "-ious" (CCVC-ious). We ensured that the structure of all pseudowords followed English phonotactics. The total number of stimuli (twelve) was chosen based on pilot testing, which indicated that this was a manageable memory load for participants under time constraints, while still exceeding the capacity limits of short-term memory [51].

As outlined above, all stimuli were designed to be either "appealing", "neutral" or "unappealing", by using consonants and vowels that ranked differently in Crystal's [8] phonaesthetic hierarchy to fill the slots in the above-described structure. In Crystal's analysis, so-called "N values" were used to measure the number of occurrences of specific phonemes in the set of words that were regarded as appealing by his study participants. We utilized phonemes with higher N values for stimuli of the appealing condition, and, conversely, phonemes with lower N values for stimuli of the unappealing condition. For the neutral condition, we used phonemes with intermediate N-values. Rather than applying strict cutoffs for these categories, we assigned stimuli to conditions based on the average N value of all phonemes within each pseudoword. While we aimed to select phonemes from the upper, middle, or lower range of Crystal's ranking, the categorization was constrained by the need to construct phonotactically valid stimuli. As a result, some phonemes appear in more than one condition, but the overall average N values differed consistently across conditions (Table 1). To present the stimuli visually, each stimulus was assigned a matching written form.

Here, we illustrate the stimulus design process, using the example of the pseudoword "clisious." The consonants /k/ and /l/ were picked from the top of Crystal's [8] list to form the first cluster of the stimulus. The vowel /ɪ/, also on top of Crystal's hierarchy, was selected to complete the first syllable. Another highly rated consonant, /s/, and the suffix /ɪəs/ were added to complete the stimulus. Finally, the written form "clisious" was assigned to represent the spoken, phonemic form of the stimulus created earlier. The list of all stimuli and their structure is outlined in Table 1. A microphone (Tonor Q9) was used to record the pseudowords, with the initial syllable being stressed in each stimulus. To keep variation in voice quality, speech tempo and intonation to a minimum, all stimuli were recorded by the same male speaker in a single recording session. All recordings were checked for clipping and inaccuracies in Audacity [52], and amplified and normalized to constant volume.

## 2.4. Analyses

### 2.4.1. Aesthetic appeal.
To test whether the aesthetic perception of the pseudowords was influenced by the phonemes that they contained, i.e., whether words in our three aesthetic conditions would receive different ratings,

**Table 1. List of stimuli used in the experiment.**

| Condition | Phonemic Structure | | | | | Written Form |
|---|---|---|---|---|---|---|
| | N Value | | | | | Average N Value |
| APPEALING | k | l | ɪ | s | ɪəs | clisious |
| | 28 | 59 | 49 | 35 | – | 42.75 |
| | s | n | e | l | ɪəs | snelious |
| | 35 | 33 | 24 | 59 | – | 37.75 |
| | s | l | iː | m | ɪəs | sleemious |
| | 35 | 59 | 14 | 40 | – | 37.00 |
| | s | m | æ | n | ɪəs | smanious |
| | 35 | 40 | 24 | 33 | – | 33.00 |
| NEUTRAL | k | r | iː | t | ɪəs | creetious |
| | 28 | 29 | 14 | 26 | – | 24.25 |
| | k | r | aɪ | d | ɪəs | kridious |
| | 28 | 29 | 13 | 24 | – | 23.5 |
| | d | r | aɪ | k | ɪəs | drikious |
| | 24 | 29 | 13 | 28 | – | 23.5 |
| | d | r | ʌ | t | ɪəs | drutious |
| | 24 | 29 | 12 | 26 | – | 22.75 |
| UNAPPEALING | k | r | aʊ | w | ɪəs | krauious |
| | 28 | 29 | 1 | 8 | – | 16.5 |
| | t | w | uː | h | ɪəs | twuhious |
| | 26 | 8 | 7 | 6 | – | 11.75 |
| | d | w | aʊ | g | ɪəs | dwougious |
| | 24 | 8 | 7 | 7 | – | 11.5 |
| | g | r | uː | h | ɪəs | gruhious |
| | 7 | 29 | 1 | 6 | – | 10.75 |

The first column shows the experimental condition. The second column shows the phonemic make-up of the pseudo-words based on the N values (corresponding to appeal) assigned to each phoneme by Crystal [8]. The third column shows the written form of the pseudo-words and their average N values, based on the N values of the individual phonemes.

we computed the mean values and their 95% confidence intervals (CIs) for each condition. Non-overlapping CIs are interpreted as indicating significant differences between conditions [53].

Additionally, we employed a Cumulative Link Mixed Model (CLMM) [54–56], using Laplace approximation [57,58]. To account for the ordinal nature of the Likert scale ratings and as CLMMs are specifically designed for analyzing ordered categorical data [56], we employed a CLMM instead of a linear mixed-effects model. In this model, *aesthetic condition,* which was the predictor of our main interest, was included as a fixed effect. Additionally, we included *trial number* as a fixed effect. It served as a control predictor to ensure that ratings were not influenced by the point in the experiment when a particular stimulus occurred. We also entered a random intercept of *participant* into the model. To prevent inflated type I error rates, each model included a random slope [59] of *aesthetic condition* within *participant*. An initial model that also included a random slope of *word* within *participant* did not converge. The sample size for the model was 2,400 data points (100 participants each provided 2 ratings for stimuli across 3 aesthetic conditions that were comprised of 4 pseudowords each). The neutral aesthetic condition was set as the reference level in the model.

The model was fitted in R [60], using the function *clmm* of the R-package "ordinal" [61].

We employed a likelihood ratio test to test the overall significance of the full model, comparing it to a null model comprising only the random effects and the control predictor of *trial number* (R function *anova;* [62]).

**2.4.2. Memory and recall.** Memory was assessed by counting how many pseudowords participants correctly recalled during the recall phase of the experiment. As participants occasionally made typos when recalling the pseudowords (e.g., typing "cliisious" instead of "clisious" or "grhuious" instead of "gruhious"), we implemented a flexible criterion to account for minor deviations. To achieve this, we applied the Jaro-Winkler distance [63–65], which calculates a similarity score between two strings based on shared characters, their order, and the number of transpositions (swapped characters). The score ranges from 0 (completely dissimilar) to 1 (exact match). We considered a response correct if its Jaro-Winkled similarity to the original stimulus was greater than or equal to 0.85, which corresponds roughly to a difference of 1–2 characters. For example, "cliisious" (extra "i") and "grhuious" (swapped letters "u" and "h") were accepted as matches for "clisious" and "gruhious", respectively. In contrast, responses like "glishous" or "corvius", which are substantially different from any original stimulus, were excluded. By doing so, we ensured that responses that were intuitive matches to the original stimuli were counted, while responses with significant differences were excluded.

To test if the participants' word recall was influenced by the phonemes that the words contained, i.e., by the aesthetic condition that the words belonged to, we fitted a Generalized Linear Mixed Effects Model [66] with a logit link function [67]. The model included a fixed effect of *aesthetic condition*, a random intercept effect of *participant* and a random slope of *aesthetic condition* within *participant* [59]. We set the appealing aesthetic condition as the reference level in the model. To test if the participants' recall of words was correlated with their own ratings of those words, we fitted another Generalized Linear Mixed Effects Model with a logit link function that included a fixed effect of the word's *rating* (each participant's mean rating of each word from rounds 1 and 2), a random intercept effect of *participant* and a random slope of *rating* within *participant*. The dependent variable in both models was whether the participants had correctly recalled the respective word (with the levels "yes" and "no"). The sample size of both models was 1200 data points (100 individuals could recall 4 words in 3 aesthetic conditions each), 523 of which were correctly recalled words.

The models were calculated in R [60], using the function *glmer* of the R-package *lme4* (version 1.1–33; [68]). To test the overall significance of *aesthetic condition* or *rating*, respectively, on word recall, we employed likelihood ratio tests to compare our full models to null models that did not include *condition* or *rating*, respectively, but only the random intercept of *participant* and the random slope of *condition* or *rating*, respectively, within *participant* (R function *anova*; [62]). To evaluate the goodness-of-fit of our models, we computed both marginal and conditional R2 values for our full models [69]. The marginal R2 ($R2_m$) reflects the variance accounted for solely by the fixed effects, while the conditional $R2(R2_c)$ represents the variance explained by both the fixed and the random effects combined. Thus, these measures give an indication of the model's overall effect size. We calculated both R2 values using the *r.squaredGLMM* function from the *MuMIn* package, using the "theoretical" method [70]. We preregistered all hypotheses, study protocols, and analyses, including provisional R files, on the Open Science Framework (Preregistration date: 27 June, 2023; https://osf.io/npe2g/overview).

## 3. Results

### 3.1. Aesthetic appeal

Overall, the full model (Table 2) differed significantly from the null model, which shows that the aesthetic condition, i.e., the phonemes and phonotactic patterns that a word consisted of, influenced how aesthetically pleasing participants rated the words (likelihood ratio test: $\chi2 = 38.59$, $df = 2$, $p < 0.001$; Fig 2A). However, in contrast to our predictions, words in the neutral condition received higher ratings (mean rating: 4.17 ± CI 0.11) than words in the appealing (mean rating: 3.60 ± CI 0.13) and unappealing (mean rating: 3.56 ± CI 0.12) conditions (see model results in Table 2 and non-overlapping 95% confidence intervals in Fig 2B). There was no significant difference in the ratings of the words in the appealing and unappealing condition (see model results in Table 2 and overlapping 95% confidence intervals in Fig 2B). Thus, our results did not replicate Crystal's [8] rankings of the aesthetic appeal of phonemes that would have predicted the aesthetically appealing condition to get the highest ratings. This pattern is also reflected by the ratings for individual words, which resemble an inverted U-shaped curve (Fig 2C), with the most and least appealing words according to Crystal's ranking

**Table 2. Results of the Cumulative Link Mixed Model exploring the effects of aesthetic condition on the ratings of participants.**

| Model coefficients | Estimate | SE | z | p | |
|---|---|---|---|---|---|
| Condition_appealing | −0.76 | 0.19 | −4.05 | <0.001 | |
| Condition_unappealing | −0.80 | 0.13 | −6.15 | <0.001 | |
| Trial number | −0.01 | 0.01 | −1.56 | 0.12 | |
| | | | | | |
| **Threshold coefficients** | **Estimate** | **SE** | **z** | | |
| 1\|2 | −3.53 | 0.18 | −19.28 | | |
| 2\|3 | −1.98 | 0.17 | −11.65 | | |
| 3\|4 | −0.85 | 0.17 | −5.13 | | |
| 4\|5 | 0.28 | 0.17 | 1.72 | | |
| 5\|6 | 1.36 | 0.17 | 8.05 | | |
| 6\|7 | 2.74 | 0.18 | 15.08 | | |
| | | | | | |
| **Random effects** | **Term** | **Variance** | **SD** | **Corr** | **Corr** |
| Participant | Intercept | 1.77 | 1.33 | | |
| | Condition_appealing | 2.68 | 1.64 | −0.28 | |
| | Condition_unappealing | 0.88 | 0.94 | −0.31 | 0.12 |

Model formula: Rating ~ Condition + Trial number + (Condition|Participant). The table reports estimated model coefficients (Estimate), standard errors (SE), z-values (z) and p-values (p) of the fixed effects, estimates, standard errors and z-values of the threshold coefficients, as well as variance, standard deviation (SD) and correlation coefficients (Corr) of the random effects. In cumulative link mixed models, threshold coefficients define the boundaries between adjacent levels of the ordinal outcome on an underlying continuous (latent) scale. Each threshold (e.g., "1|2", "2|3") marks the point on this latent scale where the probability of selecting a particular response category shifts from one to the next. For instance, the threshold labeled "1|2" represents the point separating categories 1 and 2. These thresholds are estimated along with the model coefficients and help to translate the linear predictor into probabilities for each ordinal category [71,72].

receiving the lowest ratings, and neutral words receiving the highest ratings. The only exception to this curve is the word *clisious*, which was designed to be the most appealing, and received higher ratings than expected based on the inverted U-shaped pattern. There was no significant effect of *trial number* on the ratings (Table 2), indicating that experiment progress did not lead to a consistent change of the participants' ratings.

A Spearman rank correlation analysis was performed to examine the association between the ratings obtained in round 1 and round 2 of the rating task. This revealed a statistically significant positive correlation (rho = 0.66, p < 0.001), indicating that participants who gave high ratings for particular words in rating round 1 were likely to also give high ratings for those words in rating round 2 (Fig 3).

### 3.2. Memory and recall

The comparison of the full and the null model indicated an effect of aesthetic condition on word recall (likelihood ratio test: $\chi 2 = 29.91$, df = 2, p < 0.001; effect size for the full model: $R2_m = 0.03$; $R2_c = 0.22$). More precisely, the full model (Table 3) showed that in the appealing condition (average percentage of recalled words per participant: 53.25 ± CI 5.63%; Fig 4A, 4B), participants recalled more words than in the neutral (average percentage of recalled words per participant: 41.5 ± CI 5.29%) and unappealing conditions (mean percentage of recalled words per participant: 36.00 ± CI 6.04%). This is also reflected in the percentage of participants remembering the individual words of each condition (Fig 4C, with the pseudowords "smanious" and "creetious" showing slight deviations from the general pattern within their respective conditions).

Our second generalized linear mixed effects model, which tested the effect of the participants' actual ratings on word recall, revealed a (non-significant) tendency for the ratings to correlate with word recall (likelihood ratio test: $\chi 2 = 3.51$,

**A: Violinplots of ratings**

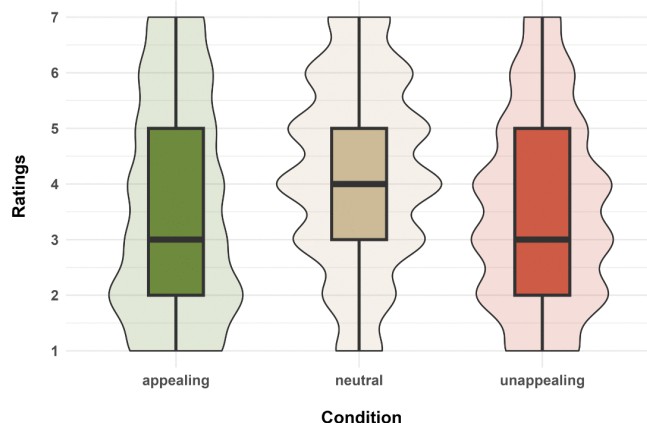

**B: Mean ratings with 95% confidence intervals**

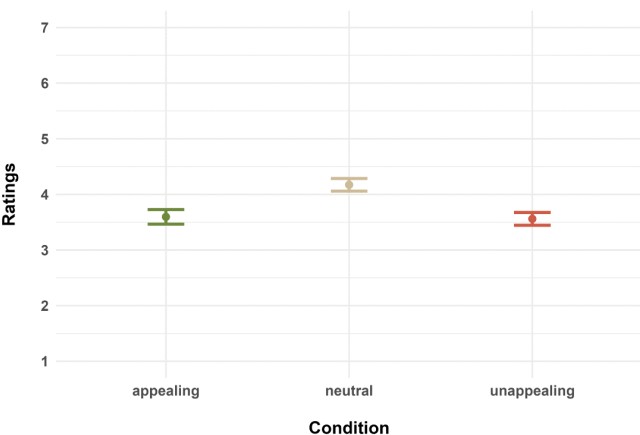

**C: Mean ratings of words with 95% confidence intervals**

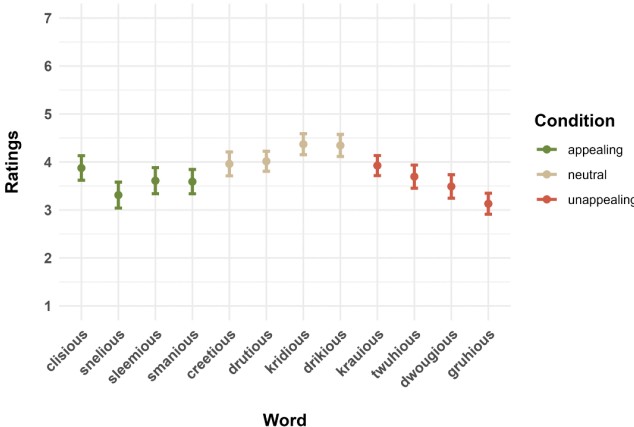

**Fig 2. Beauty ratings of words that included either appealing, neutral or unappealing phonemes (according to Crystal [8]).** A. Boxes depict medians and quartiles, and whiskers depict minimum and maximum values. Violin shapes around the boxes show the distributions of the ratings. The

width of the violin shapes at a given y coordinate corresponds to the number of ratings in this region. B. Dots denote mean values and error bars 95% confidence intervals.. C. Mean values and 95% confidence intervals of the ratings of the individual pseudowords. Mean ratings resemble an inverted U-shaped curve.

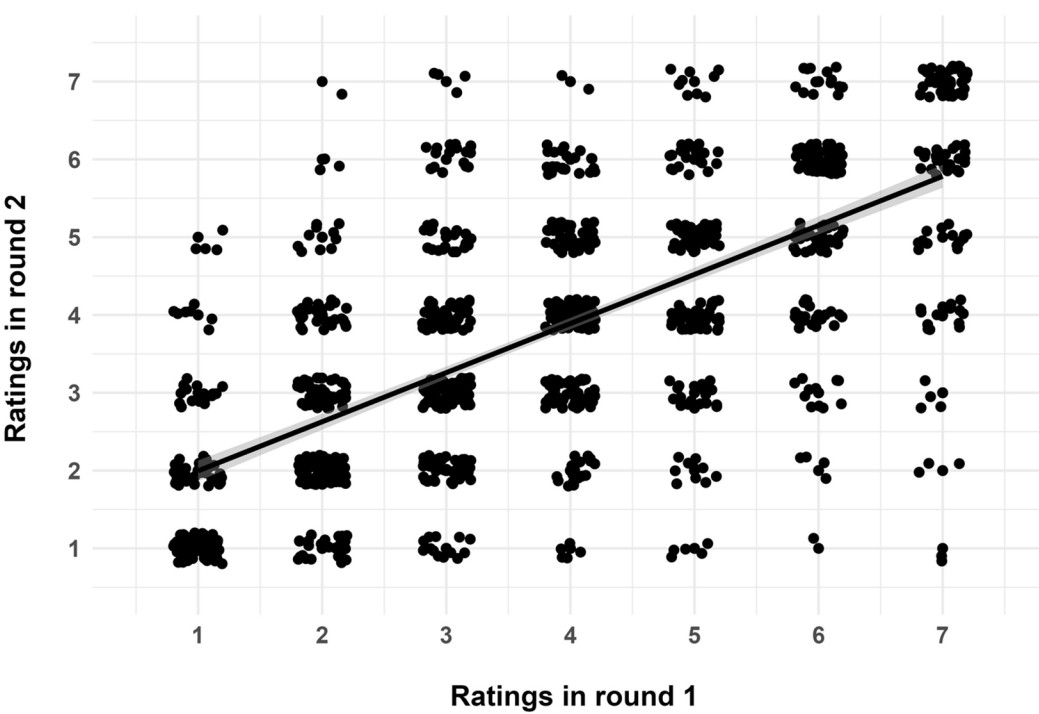

**Fig 3. Scatterplot illustrating the correlation between ratings in rating round 1 and rating round 2 of the experiment.** Each point represents an individual participant, with the x-axis corresponding to ratings in round 1 and the y-axis to ratings in round 2. Gray areas around the regression line denote 95% confidence regions.

**Table 3. Results of the generalized linear mixed effects model exploring the effects of aesthetic condition on word recall.**

| Full Model | Estimate | SE | z | p | |
|---|---|---|---|---|---|
| Intercept | 0.15 | 0.13 | 1.15 | | |
| Condition_neutral | −0.53 | 0.15 | −3.47 | <0.001 | |
| Condition_unappealing | −0.87 | 0.17 | −5.26 | <0.001 | |
| | | | | | |
| **Random effects** | **Term** | **Variance** | **SD** | **Corr** | **Corr** |
| Participant | Intercept | 0.62 | 0.79 | | |
| | Condition_neutral | 0.02 | 0.14 | −1.00 | |
| | Condition_unappealing | 0.12 | 0.34 | 1.00 | −1.00 |

Model formula: Words_recalled ~ Condition + (1 + Condition|Participant). The table reports estimated model coefficients (Estimate), standard errors (SE), z-values and p-values (p).

**A: Word recall**

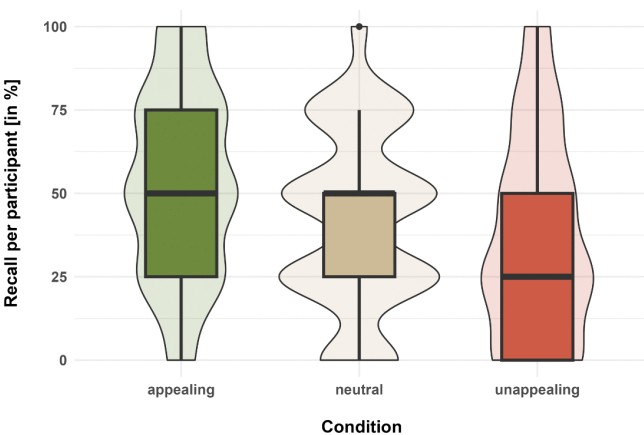

**B: Word recall**

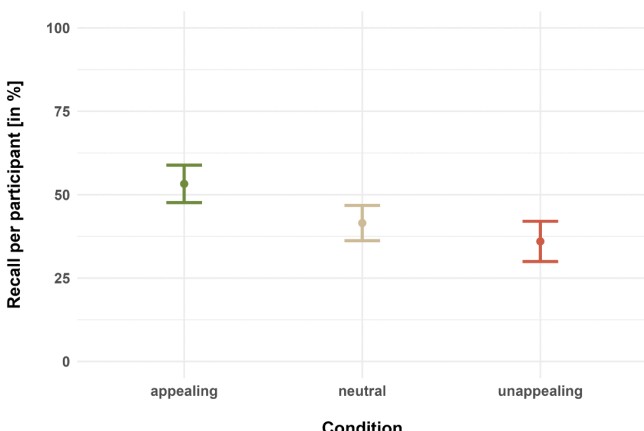

**C: Word recall of individual words**

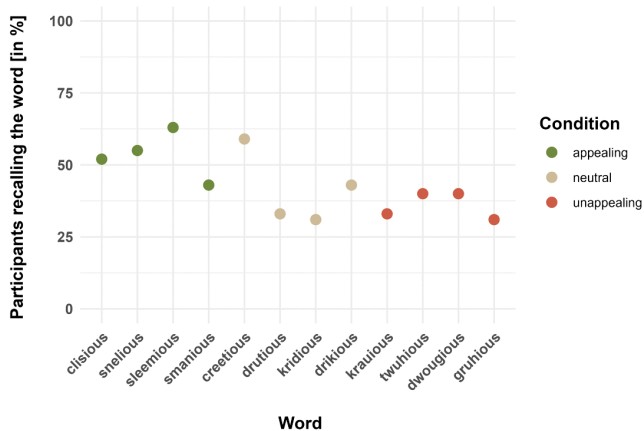

**Fig 4. Per-participant-recall of words that included either appealing, neutral or unappealing phonemes (according to Crystal [8]; in percent).**
A. Boxes depict medians and quartiles, and whiskers depict minimum and maximum values. Violin shapes around the boxes show the distributions of

the percentages of recalled words per participant. The width of the violin shapes at a given y coordinate corresponds to the number of ratings in this region. B. Dots denote mean values and error bars 95% confidence intervals. C. Percentage of participants recalling the individual pseudowords.

df = 1, p = 0.061; effect size for the full model: $R2_m = 0.01$; $R2_c = 0.18$): participants tended to be more likely to recall those words that they also rated more highly (Table 4). Also, on average, recalled words received significantly higher ratings (3.93 ± CI 0.11) than non-recalled words (3.66 ± 0.09; Fig 5, see non-overlapping confidence intervals).

## 4. Discussion

This study demonstrates that words with different phonemic and phonotactic compositions are perceived differently in terms of aesthetic appeal and that this perceived appeal correlates with word memory. However, not all of our results aligned with our specific predictions.

### 4.1. Ratings of aesthetic appeal

Contrary to our expectations, words that we had designed to be appealing based on the criteria proposed by Crystal [8] were not rated as the most appealing by our participants. Instead, they received similar ratings as words designed to be unappealing. Interestingly, words designed to have intermediate appeal received the highest ratings. A closer examination of the individual stimuli revealed, overall, an inverted U-shaped pattern in the ratings. As a result, we were unable to replicate Crystal's criteria for designing appealing words.

Several factors may explain these findings. First, Crystal's [8] rankings of phonemes, which we used to design our stimuli and experimental conditions, were based on anecdotal reports and observations. Moreover, Crystal focused on existing English words with semantic meaning, which likely influenced the perception of their appeal. In contrast, our study employed an artificial language, eliminating semantic associations and encouraging participants to focus more directly on the intrinsic appeal of speech sounds and phonotactic patterns. This methodological difference may have contributed to the divergence in results.

Secondly, the phonemic composition of the words in Crystal's [8] study might have interacted with other factors, such as syllable count or word stress, which may have influenced the perceptions of appeal but were not described in detail in Crystal's work. We opted for a controlled design with constant syllable counts and word stress. It is possible that interactions between individual speech sounds, their combinations, and other factors influenced the ratings, leading to the fact that Crystal's ranking diverged from our findings. Potentially, for the design of stimuli in future studies on aesthetic appeal, one could draw on the more recent and methodologically more controlled work on valence discussed in Section 2, e.g., [7,23,25,28]. Even though these studies do not directly focus on appeal and only partially align with Crystal's results, they may nonetheless provide a more valuable basis for developing stimuli.

**Table 4. Results of the generalized linear mixed model exploring the effects of rating on word recall.**

| Full Model | Estimate | SE | z | p |
|---|---|---|---|---|
| Intercept | −0.65 | 0.21 | −3.12 | |
| Rating | 0.09 | 0.05 | 1.90 | 0.058 |
| | | | | |
| **Random effects** | **Term** | **Variance** | **SD** | **Corr** |
| Participant | Intercept | 0.97 | 0.99 | |
| | Mean rating | 0.04 | 0.19 | −0.62 |

Model formula: Words_recalled ~ Mean_rating + (1 + Mean_rating|Participant). The table reports estimated model coefficients (Estimate), standard errors (SE), z-values and p-values (p).

**Fig 5. Mean ratings of recalled and non-recalled words.** Dots denote mean values and error bars 95% confidence intervals.

Additionally, our results align with the inverted U-curve theory of aesthetic preference, which links appeal to familiarity and suggests that moderate levels of familiarity tend to be most appealing [73,74]. According to this theory, highly frequent stimuli may lead to overstimulation and boredom, while rare or unfamiliar items may evoke uncertainty and discomfort. As a result, both extremes can reduce appeal, whereas stimuli that strike a balance between novelty and familiarity are often preferred. Because our stimulus design was based on Crystal's [8] rankings of appeal, we were unable to fully control for the occurrence frequency of phonemes in the participants' first language, English. Consequently, many of the phonemes that received high ratings in Crystal's study and were therefore placed in our "appealing" condition (e.g.,/n/,/s/,/l/), are also highly frequent in the English language [8,75,76]. This may partially account for relatively low ratings of some items in the "appealing" condition and the high ratings of those in the "neutral" condition, which included phonemes moderate in frequency. Still, familiarity alone cannot fully account for our results, as Crystal's rankings of appeal and occurrence frequency do not align perfectly. For instance,/ð/, which is relatively frequent in English (as indicated in Crystal's phoneme frequency list), is ranked as having very low aesthetic appeal in his observations.

To more closely examine the potential confounding effect of occurrence frequency and, by extension, familiarity, we conducted additional exploratory analyses (see Supporting information). Rather than relying solely on absolute phoneme frequencies (as discussed above and in [8]), we calculated positional segment frequencies and biphone frequencies for all pseudowords [77]. These measures are more fine-grained because they consider not only how often a sound occurs, but also its frequency at a particular position within a word and the likelihood of specific sound combinations. The analyses showed that words in the neutral condition contained the most frequent segments and biphones, which would fit the mere-exposure hypothesis [78] that more familiar patterns tend to be preferred. However, neither positional segment

frequencies nor biphone frequencies significantly correlated with ratings. When included as predictors in our statistical models, these measures also had no significant effect on perceived appeal. This suggests that the effects of appeal condition persisted even when controlling for phonotactic probability, and that aesthetic judgments may involve factors beyond simple familiarity. Nevertheless, future research should explicitly manipulate occurrence frequency and phonotactic probability to clarify their role in shaping perceptions of linguistic appeal. Such research could also take lexical neighborhoods into account [79,80] and examine appeal ratings of real-language neighbor words (e.g., *creepy* or *creaky* for *creetious*, or *gruesome* for *gruhious*). This would help to determine whether the number of neighbor words and their semantic content, cf. [81], can influence the perceived appeal of sounds and sound combinations.Finally, on a more speculative note, another possible explanation for our findings relates to broader cultural trends. Aesthetic preferences may also fluctuate over time [82], and it is possible that linguistic aesthetic preferences have shifted since the 1990s when Crystal conducted his study. Such shifts could further explain differences between our findings and his observations.

Irrespective of the ratings across conditions, our study revealed that participants' ratings were consistent across the two rounds of rating, indicating that within individuals, perceptions of word appeal are stable rather than arbitrary. This aligns with research from other modalities, such as music, where listeners demonstrate a high level of internal stability and consistency in their ratings [83]. The fact that we achieved similar results for linguistic stimuli suggests that a systematic investigation of words' aesthetic appeal based on ratings is feasible, reflects true aesthetic preferences, and is not merely reliant on spontaneous impulses.

## 4.2. Memory and recall

Regarding the memorization of words with different levels of appeal, we found that the words from our appealing condition were recalled most often, with average recall rates slightly above 50%. Note that recall rates around 50% in free recall tasks do not reflect chance performance as they might in recognition tasks such as 2-alternative-forced-choice tasks [84]. Given the difficulty of free recall [85], especially for pseudowords [86] after paced presentation [87], this reflects a relatively strong memory performance.

While this aligns with our initial prediction that words designed to be appealing would be best remembered, it is important to interpret this finding cautiously, as these words did not receive the highest ratings in our rating task (see discussion above). Still, notably, when comparing the appeal ratings of recalled versus non-recalled words – focusing on the actual ratings rather than the conditions – we found that recalled words received significantly higher ratings than those not recalled. While this suggests a general link between perceived appeal and word memory, with higher recall rates being correlated with higher appeal, the detected effects were small. Thus, overall, there appears to be a connection between appeal and memory, but the relationship is complex and not entirely straightforward.The frequency of phonemes within a language (in this case, English) may not only influence the ratings of aesthetic appeal (as discussed above) but could also affect word memory and recall [88–90]. Typically, familiar patterns – such as those resembling common features of the languages speakers know – are more likely to be remembered, e.g., [91–95]. However, our exploratory analyses (see Supporting information) did not support this assumption. We found no significant correlation between positional segment frequency or biphone frequency and word recall. This is because words in the neutral condition had the highest positional segment and biphone frequencies, while words in the appealing condition were remembered best. Moreover, when added as predictors in our models, positional segment and biphone frequencies were not significant. Given the complex relationship between the frequency and appeal of speech sounds (as discussed above) and in turn also between appeal and memory, it would be valuable to disentangle the influence of occurrence frequency in a separate experiment. Such a study could systematically vary the exposure frequency to specific (pseudo) words and phonotactic patterns to assess how these factors independently influence both appeal and memory. Additionally, future analyses could take into account the occurrence frequencies of similar-sounding neighbor words. In principle, words with larger neighborhoods might be remembered better due to having more connections in the mental lexicon, whereas words with smaller neighborhoods

might be more distinctive and thus easier to recall. However, previous research has found mixed evidence: Ballot et al. [96] found no effect of neighborhood size on free recall, and Goh and Pisoni [97] reported similar results, with only one experiment showing an advantage for words with smaller neighborhoods, and several others revealing no effect. Future work could systematically manipulate neighborhood size to clarify its potential role in memory for pseudowords, especially in relation to aesthetic appeal.

Our findings on the perceived appeal and memorability of words align with research in other domains of aesthetic perception. For example, studies have shown that facial prototypicality enhances attractiveness, with more average or prototypical faces being rated as more appealing than distinctive ones, e.g., [98,99]. However, Sarno & Alley [100] demonstrated that distinctiveness enhances memorability, with less prototypical faces often being easier to recognize. These findings are consistent with other research that postulates that appeal and memorability are driven by different cognitive mechanisms: appeal is related to prototypicality and familiarity [101], whereas memorability relies on distinctiveness [102]. This mirrors the results of our study, where stimuli from the neutral condition were rated as most appealing but were only moderately memorable, while stimuli from the appealing condition were more memorable despite receiving lower overall appeal ratings.

One potential limitation of our memory task is that it involved written input and responses, which may have introduced visual memory components via the activation of graphemic representations. This could have interacted with participants' phonological memory and may have confounded the influence of phonaesthetic perception on word recall. We chose a written, production-based recall task to enable more precise scoring, to ensure compatibility with our digital online data collection format and to focus on deeper memory performance, cf. [87,103]. Also, a production task was used to avoid the ceiling effects observed in pilot recognition tasks. Still, future studies might consider using purely auditory-based memory tasks, such as the serial nonword recognition task, e.g., [104], which tests phonological short-term memory by acoustically presenting two sequences of pseudowords and asking whether their order is the same or different – thus avoiding graphemic influences. However, as such tasks target memory for sequences rather than individual word forms with specific phonotactic patterns, adapting them to investigate correlations with aesthetic appeal would require careful redesign, for example through auditory recognition of individual items or oral recall methods, which also minimize the influence of visual memory.

### 4.3. Outlook and conclusions

Our study offers initial insights into the relationship between the aesthetic appeal of linguistic patterns and its link to memory. However, additional research is necessary to develop a more comprehensive understanding of this connection. For instance, while our investigation focused exclusively on speech sounds, future studies could expand this scope to encompass broader linguistic patterns, such as phonotactic patterns, word stress patterns or intonation contours.

Additionally, it is difficult to disentangle aesthetic appeal from related concepts that may influence how we perceive linguistic patterns. For example, factors such as naturalness or markedness [105], which are linked to the frequency of patterns in languages that the speakers are typically exposed to, or complexity, which can manifest in cognitive processing effort, may also play critical roles in shaping perceptions of appeal (cf. [106–108] in the visual and musical domain). Future studies should aim to control or isolate these factors by manipulating the input frequency of artificial stimuli or incorporating phoneme and phonotactic frequencies more systematically into stimulus design. Additionally, experiments could explore the directionality of the causal relationship between appeal and memory to disentangle whether patterns are more memorable because they are perceived as appealing, or whether patterns are perceived as more appealing because they are more memorable.

Further studies should also consider including participants of diverse linguistic backgrounds to determine whether the aesthetic perception of speech sounds is driven by cross-linguistic cognitive constraints or shaped by cultural influences (as, e.g., Anikin et al. [10] did for whole languages). Finally, considering the multimodal nature of communication, e.g.,

[109,110], examining the appeal of gestures or signs might provide a valuable addition to the investigation of spoken and written linguistic patterns.

Although our findings only suggest a tentative link between aesthetic appeal and word memory, they nonetheless point to potential practical and theoretical implications. If future studies with larger stimulus sets confirm that appealing patterns are easier to learn and remember, these results could, on a practical level, inspire practices and applications for language teaching and marketing. This could, for example, happen by leveraging the appeal of patterns to acquire languages or promote products more effectively, e.g., [23]. On a theoretical level, a learning advantage for appealing patterns could facilitate their adoption and transmission within communities as well as across speaker generations, e.g., [111,112]. Over time, these patterns could become embedded in cultural artifacts such as literature, poetry or music, further promoting their persistence and spread. Appeal may therefore serve as a driving force in cultural language evolution, shaping the transmission and transformation of linguistic patterns over time. Thus, our study provides an initial step towards understanding the links between aesthetic appeal, memory, and cultural language evolution, paving the way for larger research avenues.

## Supporting information

**S1 Supporting information. Appendix – Exploratory analyses: the influence of phonotactic probability on aesthetic appeal and recall.**
(DOCX)

## Author contributions

**Conceptualization:** Theresa Matzinger, David Košić.

**Data curation:** Theresa Matzinger.

**Formal analysis:** Theresa Matzinger, David Košić.

**Funding acquisition:** Theresa Matzinger.

**Investigation:** Theresa Matzinger, David Košić.

**Methodology:** Theresa Matzinger, David Košić.

**Project administration:** Theresa Matzinger.

**Resources:** Theresa Matzinger.

**Supervision:** Theresa Matzinger.

**Visualization:** Theresa Matzinger, David Košić.

**Writing – original draft:** Theresa Matzinger, David Košić.

**Writing – review & editing:** Theresa Matzinger, David Košić.

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
