## [Decision Letter · Decision Letter 0]

8 Apr 2025

Dear Dr. Matzinger,

Thank you for submitting your manuscript to PLOS ONE. After careful consideration, we feel that it has merit but does not fully meet PLOS ONE’s publication criteria as it currently stands. Therefore, we invite you to submit a revised version of the manuscript that addresses the points raised during the review process.

We look forward to receiving your revised manuscript.

Kind regards,

Li-Hsin Ning

Academic Editor

PLOS ONE

 [This work was supported by a Disruptive Innovation Grant from the Austrian Academy of Sciences and the Austrian Science Fund (grant number: DI_2023-108_MATZINGER_BEALP) awarded to Theresa Matzinger.]. 

Additional Editor Comments:

This is an interesting study that explores the interaction between perceived appeal and memorability. I have several concerns that I hope can be addressed.

Page 6, Line 16: Please specify the intertrial interval. How much time were participants given to repeat each stimulus? Was there a break between each experimental phase?

Page 7, Line 4: In the stimulus recall phase, what was the time limit for participant responses?

Page 15, Table 2: What do the threshold coefficients represent? Please clarify the meaning of labels such as "1|2" and "2|3."

Page 19, Figure 2: The so-called “appealing” condition was not rated as appealing by participants, whereas the neutral condition was. Could this be due to the use of less frequent consonant clusters in English (e.g., sn- and sm-) in the appealing condition?

Page 24, Figure 4: Although the appealing condition appeared to be more memorable, recall percentage hovered around 50%, which is at chance level. This raises questions about whether the stimuli were genuinely memorable. I would encourage a more in-depth discussion of this finding.

Additional point: Is using only four stimuli per condition sufficient to assess memorability? Some justification or discussion of this design choice would be helpful.

Reviewers' comments:

Reviewer's Responses to Questions

**Comments to the Author**

1. Is the manuscript technically sound, and do the data support the conclusions?

Reviewer #1: No

Reviewer #2: Partly

Reviewer #3: Yes

2. Has the statistical analysis been performed appropriately and rigorously?

Reviewer #1: Yes

Reviewer #2: Yes

Reviewer #3: Yes

3. Have the authors made all data underlying the findings in their manuscript fully available?

Reviewer #1: Yes

Reviewer #2: Yes

Reviewer #3: Yes

4. Is the manuscript presented in an intelligible fashion and written in standard English?

Reviewer #1: Yes

Reviewer #2: Yes

Reviewer #3: Yes

Reviewer #1: Summary and Assessment

This study assumes that the phonological composition affects the aesthetic rating and the memory of English-like nonwords and tested the hypothesis in a nonword memorization and mapping experiment. The authors found that, contrary to their hypothesis that phonemes rated as more appealing in a previous hypothesis would lead to a higher nonword aesthetic rating, nonwords with more neutral phonemes were rated more aesthetic than those with putatively more and less appealing phonemes. The performance of the memory task was in line with the hypothesis: Nonwords with more appealing phonemes were recalled better than those with putatively neutral and less appealing phonemes. While I do believe in the connection between diverse cognitive aspects and sounds, given the growing body of sound symbolism, the theoretical basis and methodological design have critical issues, which prevent the study from validating the link between sounds and aesthetics with its experimental results, as explained below. Since the issues cannot be addressed simply with a revision, I recommend a rejection to the editor(s) of the journal.

Major comments

The most critical issue in the study is the lack of a sound theory of how aesthetics is manifested in sounds or sound combinations. The brief literature review on p.3 only scratches the surface; for instance, “more appealing words contained fewer harsh and abrupt sounds” (line 9), “fricatives were consistently associated with positive meanings” (lines 25-26), and “words containing /i/ were matched with positively connotated people” (lines 29-30). None of these reviews explain why and how these links could be established, so the study does not begin with a plausible hypothesis and a clear research target. This vagueness might be due to the lack of a clear definition of aesthetics (or appeal) per se. The absence of the theory also potentially undermined data collection. On p.7, the authors explain that participants were not instructed to follow any definition of “appeal”, which was thus subject to individual interpretations (lines 14-16). In this case, how can we be sure that the subjects made qualitative similar appeal ratings based on the composition of sounds? The authors are thus encouraged to dig deeper into the (cross-linguistic) sound symbolism literature (e.g., Blais et al., 2016; Shinohara & Kawahara, 2016; Johansson et al., 2020) to develop their theory of phonaesthetics and frame their arguments based on this theory.

The design of the experiment is also problematic in several ways. First of all, while the authors presented a clear hypothesis of why more appealing nonwords should be memorized and recalled better (p.4-5), nonword processing and recall are highly complicated and influenced by a wide range of linguistic factors. For instance, the literature (e.g., Gathercole et al., 1999; Roodenrys & Hinton, 2002; Vitevitch et al., 1997) has shown the influences of phonotactic probability (PP; e.g., the transitional probability of bigrams) on nonword recall. I do not see any attempt to de-confound the effects of these factors from the predicted influence of aesthetics in the study, which renders any conclusions related to phonaesthetics based on the experimental results questionable. In fact, using the online PP calculator for English available at https://sll.ku.edu/phonotactic-probability, I found that the “appealing” and “neutral” nonwords have a higher PP than “unappealing” nonwords, which can also explain the recall performance without recourse to any assumption of phonaesthetics. The incorporation of the visual modality may have further complicated the issue since the subjects not only had to process auditory forms but also to form the correspondence between sounds and spellings. Aside from the potential additional task demands other than making aesthetic judgments, the grapheme-to-phoneme correspondence was not properly designed in the stimuli. The “appealing” nonwords had no ambiguous or rare grapheme-to-phoneme mapping immediately before the pseudo suffix [ɪous]; in the “neutral” words for instance, as the spelling tious could map to [ʃəs] as in real English words like cautious, an ambiguity that could lead to a more difficult mapping between the spelling and the intended auditory form, which in turn led to a poorer recall performance of the “neutral” nonwords compared to “appealing” nonwords. Finally, nonword memory should have been directly assessed with the oral recall of target nonwords as in the literature rather than via typing, which is undesirably influenced by orthographic transitional probabilities and the ambiguity in grapheme-to-phoneme correspondence, as explained above.

Minor comments

The authors refer to their experiment as artificial language learning, which is not accurate. An artificial language/grammar learning experiment tests hypotheses of whether hidden structural regularities (e.g., phonological assimilation, case marking, center-embedded structure) are acquired as abstract generalizations after brief exposure to learning input and extended to novel forms. In the authors’ experiment, while the nonwords were also artificial, the subjects only tried to memorize and recall individual sound-spelling pairings. No acquisition of abstract linguistic patterns was involved.

References (not cited in the authors’ work)

Blasi, D. E., Wichmann, S., Hammarström, H., Stadler, P. F., & Christiansen, M. H. (2016). Sound-meaning association biases evidenced across thousands of languages. Proceedings of the National Academy of Sciences of the United States of America, 113(39), 10818–10823. https://doi.org/10.1073/pnas.1605782113

Gathercole, S. E., Frankish, C. R., Pickering, S. J., & Peaker, S. (1999). Phonotactic Influences on Short-Term Memory. Journal of Experimental Psychology: Learning Memory and Cognition, 25(1), 84–95. https://doi.org/10.1037/0278-7393.25.1.84

Johansson, N., Anikin, A., & Aseyev, N. (2020). Color sound symbolism in natural languages. Language and Cognition, 12(1), 56–83. https://doi.org/10.1017/langcog.2019.35

Roodenrys, S., & Hinton, M. (2002). Sublexical or lexical effects on serial recall of nonwords? Journal of Experimental Psychology: Learning, Memory, and Cognition, 28(1), 29–33. https://doi.org/10.1037/0278-7393.28.1.29

Shinohara, K., & Kawahara, S. (2016). A cross-linguistic study of sound symbolism: The images of size. Proceedings of the 36th Meeting of BLS 36, 396–410. https://doi.org/10.3765/bls.v36i1.3926

Vitevitch, M. S., Luce, P. A., Charles-Luce, J., & Kemmerer, D. (1997). Phonotactics and Syllable Stress: Implications for the Processing of Spoken Nonsense Words. Language and Speech, 40(1), 47–62. https://doi.org/10.1177/002383099704000103

Reviewer #2: Great idea and a very interesting paper. All the comments in the body of the document are just suggestions. Overall, I would add more literature from sound symbolism, add some clarifications and references where needed and word the discussion in a slightly more careful manner. The notes on methodology are for future studies. (see attachment for the comments)

Reviewer #3: ===========================

Comments on PONE-D-25-04238

-------------

Main comments

-------------

This study investigates relations between phonetic/phonological composition of words, aesthetic feelings, and memorizability of words. The study makes use of pseudowords that were constructed based on the list of phonemes and aesthetic appeals by Crystal (1995). The target words were rated by English native speakers with respect to the aesthetic appealing of those words, and it was also investigated which words were recalled particularly well by these participants. The intended scale of aesthetic appealing based on Crystal (1995) was not replicated, while highly-appealing words were recalled the most reliably, as expected.

The aims and the hypotheses of the study are clearly articulated, and the structure of the study is easy to follow. The experiment is well designed and executed. The description of the methods is sufficient, though some additional information might be helpful (see comments below). The analysis is also sufficient but minimal.

There are two concerns:

1. The study assumes that pseudowords are empty in meanings and therefore free from confounding effects of semantics. The assumption might not hold (Chuang et al., 2021) and therefore might have biased the results. For example (as included in a comment below), "gruhious" was intended by design and also confirmed by subjective ratings by participants to be the least appealing. The authors interpret it as effects of phonemes and their combinations. However, it seems well likely that the lowest appealing score was due to negative words that have similar sounds, such as "gruesome". This concern can be addressed by, for example, including appealing scores of some neighbor words.

2. Related to the concern above, this study does not consider so many covariates and factors. Appealing scores of neighboring words mentioned above can be one. Another would be n-gram letter (or phoneme) frequency. It could also be syllable frequency. These measures would capture how often certain combinations of letters or phones occur. Higher frequency combinations are expected to be easier to remember.

Due to the concerns mentioned above, I recommend that the authors would check effects of appealing scores of neighbor words (similarly-sounding words) and effects of phonotactics (e.g., n-gram frequency). Except for these, the study is well designed, executed, and summarized.

Below are the comments for the points mentioned in the guideline.

Reference

---------

Crystal, D. (1995). Phonaesthetically speaking. English Today, 11(2), 8–12. https://doi.org/10.1017/S026607840000818X

Criteria by the guideline

The study connects pseudowords perception, aesthetic feelings, and memory. The combination is relatively unique. Therefore, the study meets the criterion mentioned in the guideline "The study presents the results of original research".

Regarding another criterion in the guideline "Experiments, statistics, and other analyses are performed to a high technical standard and are described in sufficient detail", the study is intermediate. The experiment in this study is mostly well designed, executed, and described. But some additional details could be addressed (e.g., whether participants could take notes, or they were instructed explicitly not to). The analysis is also enough to support the main claims of the study, but it is minimal. One concern is that almost no other covariates are included in the models than the variables of interest and random effects. As the authors mention in the discussion section, sublexical frequency such as biphone/biletter frequency could be integrated. Semantic and orthographic similarity of the target pseudowords to their closest real-word neighbors might also be taken into consideration.

The study is sufficient with respect to the following criterion mentioned in the guideline: "Conclusions are presented in an appropriate fashion and are supported by the data". The conclusions in the study are supported enough by the data and observations, although there remain some concerns regarding the lack of covariates. In addition, the observations do not align well with the literature, as the authors mention in the paper. Although the authors raise several possibilities regarding the discrepancy of the present findings and the literature, they remain speculative. In that sense, the conclusions might be perceived as "weak".

This study has no problem regarding the other guideline points. The paper is written in an intelligible fashion and is written in standard English, and the study meets applicable standards for the ethics of experimentation and research integrity. Regarding the data availability, the study meets the criterion. However, R markdown files could perhaps be rendered to html or pdf for better access to the data.

-------------------

Individual comments

-------------------

*These comments are identical with those found in their corresponding locations in the attached pdf.

p.1, l.1

--------

This is a minor comment, but this study makes target words based on the ranking of "phonemes" by Crystal (1995). Therefore, it may be misleading to say "phonetic (composition)" in the title. "Phonological" or "Phonemic" instead of "phonetic" may fit better. If you keep "phonetic" in the title, it may be helpful to indicate in the body text what the authors refer to by "phonetic".

p.4, l.3-4

----------

This is also meant to be just a minor comment. But this study does not discuss aesthetic perception of individual phonemes. This study rather focuses on pseudo-"words". So, this portion of text is good in the sense that it makes clear a gap in the literature but may not be so relevant to the present study. How about, for example, pointing out here that the literature discussed in this paragraph does not investigate relations between aesthetic perception and memory?

p.7, l.20

---------

Was it made sure that participants did not take notes during the experiment? If not, you could check the distribution of the performances among participants. Outliers (too good performance) may be a sign of taking notes.

p.8, l.6-8

----------

Psuedowords may not be "empty" in semantics (see, e.g., Chuang et al., 2021). Pseudowords may be influenced by existing real words by partial overlaps. For example, at the bottom of the material list is "gruhious". I agree that this word sounds particularly unappealing. I think it is because of the similar-sounding word "gruesome". The first few phonemes of "gruhious" perhaps evoke the meaning/feeling of "gruesome", and therefore "gruhious" may be felt unappealing.

To see if/how much neighbor real words affect the perception of aesthetic feelings for pseudowords, you could, for example, identify top 5 real words that are closest in sound to the target word (e.g., gruhious), calculate the mean value of aesthetic appeal, and put it in the models.

This comment is not intended to criticize the study, just to be clear. Some pseudowords are rated more or less appealing (and it affects memory) through shared phones with other existing words. That is in line with Crystal (1995), in which they rated aesthetic feelings of phonemes based on which words they occur in.

Chuang, Y.-Y., Vollmer, M. L., Shafaei-Bajestan, E., Gahl, S., Hendrix, P., & Baayen, R. H. (2021). The processing of pseudoword form and meaning in production and comprehension: A computational modeling approach using linear discriminative learning. Behavior Research Methods, 53, 945–976. https://doi.org/10.3758/s13428-020-01356-w

p.8, l.17-20

------------

Can you add how you defined "higher", "lower", and "intermediate"? It is mentioned later, to describe an example, that you took the most appealing phonemes from the Crystal's study to form "clisious". I could infer that you probably took the 2nd most, 3rd most, ... appealing phonemes from the Crystal's study. But, it would be nice if it were spelled out in the text. In addition, it is not clear how you chose "intermediate" from the list of phonemes of the Crystal's study.

p.12, l.14

----------

A few examples may help here to explain the Jaro-Winkler distance. What misspelling was accepted, while what misspelling was excluded?

p.14, l.3-4

-----------

A summary of the models fitted in the study may be helpful here. You could even add the model formulas in the R (glmer) syntax: such as "Rating ~ (1+Participant | AesCondition) + ... ".

p.14, l.10

----------

"as" may be a typo.

p.15, l.7

---------

A brief mention of conceptual interpretations of threshold coefficients might be helpful for those who are not familiar with an ordinal regression. It is not so well known as fixed effects and random effects.

p.19, l.2

---------

Figure 2 contains three figures. You could split them into separate figures with a shorter caption. Same for Figure 4.

p.19, l.7-8

-----------

See the comment on Figure 5.

p.21, l.18

----------

This model contained random effects, right? It would be helpful to include random effects in the summary table just for better clarity.

p.24, l.7-9

-----------

See the comment on Figure 5.

p.25, l.10

----------

It would probably be helpful to include random effects in this table as well, just for better clarity.

p.26, l.4 (Figure 5)

This text is included in every figure in which confidence intervals are drawn. It might be redundant. You could mention this only once in the body text when a figure with confidence intervals is mentioned for the first time and drop the text for the captions of the figures.

p.32, l.7

---------

To fill in?

**Do you want your identity to be public for this peer review?** For information about this choice, including consent withdrawal, please see our Privacy Policy

Reviewer #1: No

Reviewer #2: **Yes: ** Vita V Kogan

Reviewer #3: **Yes: ** Motoki Saito

---

## [Author Response · Author response to Decision Letter 1]

8 Sep 2025

Dear Li-Hsin Ning, dear editorial team,

We sincerely thank you for evaluating our manuscript “Phonemic composition influences words’ aesthetic appeal and memorability”, and for the invitation to revise it.

We are especially grateful for your and the reviewers’ many insightful remarks and very detailed helpful suggestions. We have carefully considered all comments, and believe that we have fully addressed your and the reviewers’ requests in this revised submission. We strongly believe that – thanks to your and the reviewers’ comments – the quality of the manuscript has considerably improved.

Most importantly, we have revised the introduction to provide a stronger theoretical framing, incorporating theories of sound symbolism. We have also addressed the potential confounding effect of phonotactic probabilities by adding a new analysis and have carefully toned down our claims in the discussion. All changes are reported in detail in our response to the reviewers.

We hope that our changes satisfy you and the reviewers. Please consider this revised manuscript for publication in PLOS ONE.

Many thanks!

Best,

Dr. Theresa Matzinger (for the authors)

---

## [Decision Letter · Decision Letter 1]

3 Oct 2025

Dear Dr. Matzinger,

Thank you for submitting your manuscript to PLOS ONE. After careful consideration, we feel that it has merit but does not fully meet PLOS ONE’s publication criteria as it currently stands. Therefore, we invite you to submit a revised version of the manuscript that addresses the points raised during the review process.

We look forward to receiving your revised manuscript.

Kind regards,

Li-Hsin Ning

Academic Editor

PLOS ONE

Journal Requirements:

Reviewers' comments:

Reviewer's Responses to Questions

**Comments to the Author**

Reviewer #2: All comments have been addressed

Reviewer #3: All comments have been addressed

2. Is the manuscript technically sound, and do the data support the conclusions?

Reviewer #2: Yes

Reviewer #3: Yes

3. Has the statistical analysis been performed appropriately and rigorously?

Reviewer #2: Yes

Reviewer #3: Yes

4. Have the authors made all data underlying the findings in their manuscript fully available?

Reviewer #2: Yes

Reviewer #3: Yes

5. Is the manuscript presented in an intelligible fashion and written in standard English?

Reviewer #2: Yes

Reviewer #3: Yes

Reviewer #2: Great job on very meticulous revisions! Thank you!

Abstract: Few lines are a bit confusing. “Nevertheless, pseudo-words designed to be highly appealing were recalled most frequently.” – maybe add “even though participants themselves did not rate them as highly appealing”. “Also, overall, recalled words received higher appeal ratings (add “from participants”) than non-recalled ones.”

Lit review: Minor cosmetic edits such as occasional lack of spacing, punctuations (double . .), pseudowords vs pseudo-words, some references in text are not in the alphabetic order, missing closing bracket.

p. 7: For “As a result, if speech sounds or phonotactic patterns differ in their aesthetic appeal, these differences in emotional arousal may have significant implications for our understanding of phonological short-term memory (Baddeley, 2003) and language acquisition. “ Maybe better: “As a result, if speech sounds or phonotactic patterns differ in their aesthetic appeal, these differences in emotional arousal may have significant implications for our understanding of phonological short-term memory FUNCTIONING (Baddeley, 2003) IN THE CONTEXT OF language acquisition. “

Results: In Table 2, what does the last column feature? I think the name of the column is missing.

Congratulations on an interesting paper!

Reviewer #3: This study aims to investigate the relationships between phonemes and aesthetic appeal on one hand, and between aesthetic appeal and memorization on the other hand, using pseudowords.

All the points I raised in the previous review round have been addressed. I am particularly happy to see that the effects of the experiment conditions (i.e., aesthetic appeal) remained robust and significant after controlling for phonotactic probability. I believe this additional analysis makes the main argument of this study much more convincing.

I recommend this study for publication, with only a few minor stylistic comments:

1. Several occurrences of the word "phonemic" are missing a space afterward (e.g., "phonemicand", "phonemicquality", etc.) I would recommend adding a space after "phonemic" in each of these instances.

2. In the References section, some entries are indented while others are not. I just wanted to point this out, although such formatting issues may be checked and corrected at a later stage of publication.

Apart from these minor points, this study is well-structured, well-executed, clearly summarized, and persuasively argued. Thank you very much for this interesting work. I look forward to seeing further developments in this line of research.

**Do you want your identity to be public for this peer review?** For information about this choice, including consent withdrawal, please see our Privacy Policy

Reviewer #2: **Yes:** Vita V. Kogan

Reviewer #3: **Yes:** Motoki Saito

---

## [Author Response · Author response to Decision Letter 2]

23 Oct 2025

Please see the attached document "Response to Reviewers 2".

---

## [Editor Report · Decision Letter 2]

28 Oct 2025

Phonemic composition influences words’ aesthetic appeal and memorability

PONE-D-25-04238R2

Dear Dr. Matzinger,

We’re pleased to inform you that your manuscript has been judged scientifically suitable for publication and will be formally accepted for publication once it meets all outstanding technical requirements.

Kind regards,

Li-Hsin Ning

Academic Editor

PLOS ONE
---

## [Editor Report · Acceptance letter]

PONE-D-25-04238R2

PLOS ONE

Dear Dr. Matzinger,

I'm pleased to inform you that your manuscript has been deemed suitable for publication in PLOS ONE. Congratulations! Your manuscript is now being handed over to our production team.

Kind regards,

on behalf of

Dr. Li-Hsin Ning

Academic Editor

PLOS ONE